# OpenReview forum: "UnSTAR: Unlearning with Self-Taught Anti-Sample Reasoning for LLMs"
_TMLR — Accepted by TMLR_

### Review · Reviewer_8JfS · 2025-04-07

**Summary Of Contributions:**

The paper considers the unlearning problem, where we have a model trained on a dataset and a subset of that dataset, called the forget set, is given with the goal of forgetting the information in the forget set (i.e., retrieving the model we would have trained if the forget set was originally included). The authors propose UnStar, a method using "anti-samples" to facilitate unlearning. At a high level, UnStar iteratively takes a question-answer pair, finds a paraphrased version of the question and a wrong/falsified answer, and fine-tunes the model on this new question-answer pair and a rationale for the incorrect answer. After this fine-tuning, if the model still outputs the correct answer, the process is repeated until the model outputs the incorrect answer. This is done multiple times per question-answer pair in the forget set, and to ensure the model does not forget anything about examples outside the forget set, the model is re-fine-tuned on the retain set (i.e., the dataset minus the forget set). In more detail, UnStar uses a number of techniques, such as filtering paraphrased questions using semantic similarity measures, using paraphrasings to bypass LLM guardrails, filtering out falsified answers similar to the correct answer, to refine these steps further.

Along with defining UnStar in detail, the authors give intuition for why UnStar improves on other methods. The main advantage of UnStar is that it attempts to decouple information in the forget set from information on the same topic that is not to be forgotten. In contrast, previous methods focused only on unlearning the forget set, which can come at the cost of forgetting information related to but not in the forget set (e.g., forgetting a person exists rather than just a detail about them). The authors give some theoretical intuition for this by framing UnStar as a gradient descent method on using policy gradients, for a reward defined jointly across the forget and retain sets.

The authors conduct experiments on some standard unlearning benchmarks, including Wikipedia Person Unlearn, Peter Parker forgetting, and the TOFU dataset. They compute a number of metrics and composite metrics for UnStar, and compare them to several other unlearning methods. These metrics include (i) Unlearning efficacy, i.e. how effectively the model forgets information (ii) model utility on unrelated tasks (iii) response quality when prompted on the forgotten information (iv) avoiding hallucinations on the forgotten information (v) robustness to attacks trying to recover the unlearned information. UnStar achieves the highest performance on (i) and (ii), and is competitive with the highest performance on metrics (iii)-(v); most other methods are not compettiive on at least 2 of these 5 metrics.

**Audience:**

Yes

**Broader Impact Concerns:**

No concerns.

**Claims And Evidence:**

Yes

**Requested Changes:**

Please address the following:
* The training cost of UnStar vs. retraining on the entire retain set, and the other unlearning approaches you compare to.
* How, if possible, to extend UnStar to general text datasets (i.e., each example is just a string of text with no clean separation into question/answer pairs. e.g. Wikipedia pages)

It would also be nice, but not needed, to answer the following:

* How does UnStar handle randomness in model outputs? e.g. it's possible that for a given question, after one iteration of fine-tuning the model on an incorrect answer, the model has a 90% of outputting the correct answer but outputs the incorrect answer and UnStar will no longer try to unlearn the correct answer, but it seems based on the experiments this doesn't happen. I assume this is partially why it uses multiple paraphrased questions per question-answer pair, as added redundancy that can decrease the likelihood of a false positive for unlearning by using more training time, but am not sure.

**Strengths And Weaknesses:**

Strengths:
* UnStar's empirical performance is quite strong, and the experiments are quite extensive and consider a large number of previously existing baselines. Overall the paper provides quite a convincing empirical argument for UnStar achieving better unlearning than the other algorithms.
* The authors identify a clear issue with past works (not decoupling the forget set and information outside the forget set in unlearning) and give a very targeted approach (anti-samples in conjunction with samples) that differs significantly prior unlearning works. More succinctly, the technique is novel and it is intuitively clear why it works.
* The paper includes detailed explanations of e.g. each step of UnStar, comparisons between UnStar and other methods, justifications for why UnStar succeeds on the retain set after unlearning, and despite the level of detail the presentation is overall accessible.

Weaknesses:
* UnStar re-fine-tunes on the entire dataset, even if the forget set is small. If this much computational power is affordable, it's not clear (i) why one should not retrain (perhaps from a pre-trained checkpoint) on the retain set instead, (ii) how UnStar compares to the other unlearning methods in the experiments in terms of computational cost.
* The paper assumes the dataset consists of question-answer pairs. While this can cover many use cases, it is unclear how to extend to more general text datasets, i.e. where the inputs are just text without question/answer labels (although perhaps this extension is not hard)

---

> ### Author Response · Authors · 2025-04-23
>
> **Requested Changes (RC) 1**
>
> While UnSTAR includes fine-tuning on the retain set, it is crucial to emphasize that the forget set typically contains only a small number of examples (e.g., facts about a specific individual or entity). Consequently, the retain set remains nearly as large as the original pretraining corpus, making retraining from scratch extremely expensive.
>
> As an illustrative benchmark, Luccioni et al. (2022) [1] report that training the BLOOM model (176B parameters) from scratch took over 118 days of compute time on 384 A100 GPUs and consumed 433 MWh of electricity—highlighting the massive environmental and financial costs involved in full retraining.
>
> By contrast, UnSTAR does not require training from scratch. It performs:
> 1. Targeted fine-tuning on a small number of anti-samples, and
> 2. A lightweight reinforcement phase on the retain set, typically over a few epochs.
>
> This process is:
> 1. Exponentially cheaper in time and resource cost than retraining,
> 2. Feasible even on modest GPU setups within a few hours to a day, depending on model size, and
> 3. Scalable to real-world deployment where forget requests may be frequent and varied.
>
> [1] Luccioni, A., Viguier, M., Ligozat, A. L., & Rozière, B. (2022). Estimating the Carbon Footprint of BLOOM, a 176B Parameter Language Model. arXiv:2211.02001
>
> **RC 2**
>
> Yes, UnSTAR can be extended to general text datasets by extracting implicit factual knowledge and converting it into structured formats that the method can operate on. The core idea is to construct anti-samples that contradict learned associations, even if those associations were not explicitly stored in QA format.
> We propose the following adaptation pipeline:
> 1. Fact Extraction: Use automated tools (e.g., OpenIE, relation extraction models) to identify factual assertions in the text, such as (Barack Obama, born in, Hawaii) from Wikipedia passages.
>
> 2. QA Conversion: Transform these facts into question-answer pairs using question generation models (e.g., “Where was Barack Obama born?” → “Hawaii”). This makes the data compatible with the QA-style anti-sample training loop in UnSTAR.
>
> 3. UnSTAR Application: Once converted, the full UnSTAR pipeline—including paraphrased prompts, falsified answers, and misleading rationales—can be applied without modification.
>
> 4. Generalization Back to Text: Even though unlearning happens in QA space, it impacts the model’s internal representations, so the effect will reflect in downstream tasks like summarization, completion, and dialogue that depend on the forgotten knowledge.
>
> However, this extension is viable primarily for reasoning-capable models, such as instruction-tuned LLMs (e.g., LLaMA-2, Mistral, GPT-3.5), where rationales and chain-of-thought logic influence output behavior. In contrast, non-reasoning models like GPT-J may not exhibit the necessary reasoning trace that UnSTAR leverages for constructing effective anti-samples. As a result, these models tend to either resist forgetting or collapse entirely when trained on incorrect samples, making UnSTAR less effective or inapplicable in such cases.
>
> In short, UnSTAR’s extension to general text depends not only on converting text to structured knowledge, but also on the model’s capacity for guided reasoning, which is central to both generating and responding to rationales.
>
> **RC 3**
>
> We appreciate the reviewer’s insightful observation — yes, the use of multiple paraphrased questions per QA pair is a key design choice in UnSTAR that directly addresses this concern.
>
> UnSTAR explicitly accounts for stochasticity in model outputs by incorporating redundancy through paraphrasing and repetition. For each fact to be unlearned, UnSTAR generates multiple paraphrased variants of the original question, each paired with a falsified answer and a misleading rationale. This redundancy plays a crucial role in two ways:
> 1. Mitigating False Positives Due to Sampling Noise:
>  If a model outputs the incorrect answer due to randomness rather than genuine forgetting (e.g., due to high entropy or low confidence), it could falsely signal success. By evaluating the model’s response across multiple paraphrased variants, we reduce the chance of such false positives. The unlearning loop continues until the model consistently fails to produce the correct answer across these variations, making forgetting more robust and reliable.
>
> 2. Stabilizing Updates Through Gradient Averaging:
>  Training on a semantically diverse batch of anti-samples reduces variance in the learning signal and ensures that forgetting generalizes beyond a specific phrasing. This aligns with policy gradient-like strategies, where broader sampling leads to more stable and effective updates.

---

> > ### Comment · Reviewer_8JfS · 2025-05-01
> >
> > Thanks for your response! Overall I believe my concerns have been addressed.
> >
> > It might still be useful in the empirical comparisons to include some measure of computational cost for the different methods (e.g. either wall-clock time or estimated FLOPs; including retraining from scratch) to providence evidence that UnSTAR is more efficient than retraining from scratch and also comparable to other unlearning methods, but I understand getting these numbers might be challenging.

---

> > > ### Author Response · Authors · 2025-05-03
> > > **Thank you!**
> > >
> > > We are glad to hear that you found our response satisfactory!
> > >
> > > Regarding empirical comparisons of time-cost of different unlearning methods: In Table 7 (Appendix  A.1), we have already shown  the wall‑clock cost of existing methods (SNAP, WAGLE, NPO) and compare with UnStar on 3 datasets: Harry Potter, Peter Parker, TOFU. Across all datasets UnSTAR is 2 × to  300 × faster than the strongest unlearning baselines.
> > >
> > > We did not list “retrain from scratch” because it is infeasible on our hardware to retrain for unlearning request. The cost of retraining may run into >90 GPU hours. Including that number would not alter the ranking but we can add an approximate cost of retraining in Table 7.

---

### Review · Reviewer_q1SN · 2025-04-09

**Summary Of Contributions:**

The paper introduces UnSTaR, a novel method for unlearning in the context of LLM question answering. Given a QA forget sample, UnSTaR hinges on generating multiple paraphrased versions of the question and corresponding incorrect answers, with model-generated justifications. These are used to finetune the model to induce forgetting of the correct answers. The model is then finetuned in a similar fashion over the retained data to preserve model utility. The method achieves impressive performance on  the presented dataset, however the method is evaluated on only 1 model (Mistral-7B) and 1 dataset (WPU).

**Audience:**

Yes

**Broader Impact Concerns:**

There are no ethical concerns that I am aware of.

**Claims And Evidence:**

No

**Requested Changes:**

Please refer to the weaknesses listed above. At minimum, I would specifically request weaknesses [2, 3, 5, 6, 7, 9, 10] be addressed in order for me to recommend publication. For example, I would like to see:

1. Support provided for the claim made in related work.
2. A restricting of the novelty claims to only those that are defensible.
3. Additional analysis of the method.
4. Clarification and formal definition of the evaluation metrics - alongside a response as to whether these can offset the limitations of ROUGE.
5. Clarification regarding weakness 7, and relevant changes if necessary.
6. Evaluation with at least one additional model.

**Strengths And Weaknesses:**

**Strengths:**
1. The method is intuitive and straightforward, it is likely to be replicable and form a solid baseline for future works.
2. The integration with STaR and use of reasoning-based anti-samples is novel and interesting.
3. Given the generic and simple construction of the method, I believe it would likely generalise to a range of question domains, models, datasets, etc.
4. The structure of the paper is nice, and presented in a (mostly) sensible order. I found the paper easy to digest and understand in one sitting.
5. The results suggest strong performance across all metrics. The ability to unlearn fine-grained information is valuable.

**Weaknesses**:
1. In general, the language and quality of written communication could be improved throughout the paper.
2. Comparison in the related work section is superficial, and there is an unsupported claim at the end of the LLM unlearning related work subsection: "However, these methods often lack the granularity required for fine-tuned control over what specific information is forgotten, which is where our approach—utilizing anti-samples—proposes a more refined solution."
3. The claim that this paper is the first to do "anti-sample" unlearning is incorrect. By the paper's own admission, the WHP method uses anti-samples for LLM unlearning. Furthermore anti-samples can be, in my opinion, viewed as adversarial samples constructed to induce forgetting. Both [1,2] construct adversarial samples to induce forgetting in unlearning, while applied to classification and not LLMs, these methods still warrant discussion in the related work. Later in this paper, you mention claim that this work is the first to "integrate anti-samples with reasoning-based unlearning in LLMs", this is (to the best of my knowledge) accurate, and is sufficient for publication. However, the broader claim of being the first to employ anti-samples is too large of a claim.
4. The method breakdown presented on page 4 doesn't explain what constitutes a correct answer. Does it have to be identical to the original answer at the token level? Does a slight misspelling or extra space in the answer constitute a now-incorrect response? What is the criteria for an answer being correct?
5.  A central hypothesis of this paper is that providing fake justifications improve the unlearning process. While this makes intuitive sense, there is little-to-no theoretical or empirical evaluation of this claim. It would be good to see either theoretical analysis or small-scale experiments to evaluate the mechanisms of the unlearning approach, and to understand precisely the influence of how these justifications impact the model.
6. The author's note they use ROUGE score, but I'm wondering how useful this is. If I ask "who is harry potter" and the correct answer is "a wizard from Hogwarts" but the LLM answers "a fictional magic user who defeats Voldemort" it clearly has not unlearnt the necessary information, but the ROUGE score would suggest it had unlearned. It's possible that the GPT privacy/quality scores address this limitation - but the author's never define any of the metrics they evaluate against. Does the privacy score address these situations?
7. The author's mention they finetune over the retain set to preserve performance. When you evaluate the model utility, are these evaluations drawn from the retain set? The premise of unlearning is that we cannot finetune/retrain over the entire dataset due to the prohibitive cost. If you finetune over the entire remaining retain set, and then only evaluate over samples that are within this retain-set, then we can't draw conclusions about whether the preserved model utility generalises to other knowledge. Are you evaluating over a hold-out set?
8. It would be good to see Figure 3 with another line denoting model utility, to see if improving the unlearning degrades retain-set performance.
9. The evaluation metrics are never formally defined. What is concretely meant by the ChatGPT privacy/quality score? Is this just Chat-GPT rating the response?  All of the metrics should be more clearly defined.
10. While I appreciate the breadth of baselines compared against, comparison on a single dataset and a single model does limit the extent to which the results have meaning. We cannot know if this method generalises at all from the current results. Other recent papers have presented results on multiple models. Evaluation on a second model, even a small one, would be appreciated.
11. Further discussion of limitations would be beneficial to the audience.


[1] Tarun, Ayush K., et al. "Fast yet effective machine unlearning." IEEE Transactions on Neural Networks and Learning Systems (2023).

[2] Chundawat, Vikram S., et al. "Zero-shot machine unlearning." IEEE Transactions on Information Forensics and Security 18 (2023): 2345-2354.

---

> ### Author Response · Authors · 2025-04-23
>
> **Requested Changes (RC) 2**
>
> We thank the reviewer for pointing this out. In the revised manuscript, we have substantially improved the LLM Unlearning subsection of the Related Work section. We now provide a more structured and detailed taxonomy of prior work, including gradient-based, adversarial, privacy-preserving, targeted, and socially-motivated approaches, along with retrieval-augmented settings and evaluation studies.
>
> For each category, we include a brief critical comparison that clarifies the scope and limitations of existing techniques. In particular, we have rephrased the final claim to be better supported by this discussion, highlighting that many existing methods operate at a coarse level—e.g., substituting full outputs, adding refusals, or injecting random labels—and may struggle with disentangling closely related facts. In contrast, our approach introduces anti-samples that enable fine-grained suppression of specific associations while retaining related knowledge.
>
> It is reproduced below for your convenience:
>
> >LLM Unlearning. Advancement in large language models has led to critical challenges, including security violations, privacy breaches of sensitive personal data, the propagation of social biases and stereotypes, the spread of misinformation such as fake news, the generation of toxic or harmful content such as hate speech or explicit material, copyright infringement of authored text or art forms, legal compliance with regulations like GDPR and CCPA, and environmental impact contributing to growing carbon footprint, raising sustain- ability concerns for the future (Bommasani et al. (2021)). Consequently, there has been a surge of interest in LLM Unlearning attempts because of their potential to improve privacy, enhance safety, and mitigate bias in large language models (Liu et al. (b), Liu et al. (a), Liu et al. (2024a), Sun et al., Farrell et al., Doshi & Stickland, Bu et al., Liu et al. (c), Choi et al. (2024a), Guo et al.).
>
> >Existing approaches can be broadly categorized into: ❶ Gradient-based methods, which apply fine-tuning or model editing to reverse prior knowledge (Wei et al.; Jin et al.; Baluta et al.; Gu et al. (2024); Jang et al. (2022); Yao et al.). While effective, these often suffer from catastrophic forgetting or collateral erasure of nearby knowledge. ❷ Adversarial and robustness-driven techniques that make the model more resistant to certain inputs or prompts (Zhao et al. (2024); Zhang et al. (2024c); Choi et al. (2024a); Yuan et al. (2024)), but may lack clarity in what is actually being removed from internal representations. ❸ Privacy-preserving strategies, which use techniques like editing, masking, or differential privacy to protect sensitive information (Jang et al. (2022); Wu et al. (2023); Lee et al. (2024); Liu et al. (2024b); Rashid et al. (2024); Kassem et al. (2023)). These methods often trade utility for generalization. ❹ Targeted unlearning, which aims to erase specific factual associations while retaining unrelated knowledge (Liu et al. (2024a); Jia et al.; Liu et al. (a); Guo et al.; Huang et al. (2024)). However, many of these operate at coarse granularity—e.g., replacing entire answers with refusals (Ishibashi & Shimodaira (2023); Choi et al. (2024b)), or training on random labels (Yao et al.), or injecting perturbed prompts (Eldan & Russinovich (2023); Liu et al. (a))—which may not fully disentangle targeted knowledge. ❺ Socially motivated approaches, which address fairness, bias, or toxic content (Patil et al. (2023); Yu et al. (2023); Liu et al. (2024c)). These typically rely on dataset-level interventions rather than controlled unlearning of specific facts. ❻ Unlearning in retrieval-augmented models (RAG), where forgetting is applied to external memory or document embeddings (Choi et al. (2024a); Lu et al. (2022); Wang et al. (2023; 2024)). ❼ Optimization and theoretical studies, which provide insight into dynamics of forgetting and bounds on knowledge removal (Zhang et al. (2024a); Scholten et al. (2024)), and ❽ Evaluation methodologies, which propose benchmarks and metrics to assess unlearning performance (Shi et al. (2024); Shumailov et al. (2024)).
>
> >While these works advance the field, most approaches lack fine-grained control over what exactly is forgot- ten—often targeting entire facts, entities, or prompt types. In contrast, our method introduces anti-samples: crafted counterfactuals that inject targeted negative training signal to selectively suppress specific associations (e.g., a profession or birthplace) without disturbing neighboring knowledge. This enables a more precise and interpretable form of unlearning, especially useful in applications requiring nuanced edits or selective retention.

---

> ### Author Response · Authors · 2025-04-23
>
> **RC 3**
>
> We agree that the broader claim of being the first to use anti-samples for unlearning is too strong and have revised our claim accordingly. As rightly pointed out, WHP does employ token-based anti-samples for LLM unlearning, and earlier works [1,2] in classification settings use adversarially generated samples to induce forgetting. We now cite and discuss these works in our related work section.
>
> That said, as also highlighted by Reviewer Jzjz, our method is conceptually distinct in that we focus on reasoning-driven anti-sample generation, going beyond simple random-labeling or token perturbation approaches. Specifically, we integrate chain-of-thought rationales to generate structured, misleading explanations for incorrect answers. This integration of reasoning with anti-sample unlearning — to our knowledge — is novel and represents a meaningful contribution in the context of unlearning for LLMs.
> We have edited this in our paper, for example in introduction:
>
> >Although token-based anti-samples have been previously introduced (e.g., Who Eldan & Russinovich (2023)), the use of reasoning-driven anti-samples—constructed with falsified answers and misleading rationales—remains novel.
>
> [1] Tarun, Ayush K., et al. "Fast yet effective machine unlearning." IEEE Transactions on Neural Networks and Learning Systems (2023).
>
> [2] Chundawat, Vikram S., et al. "Zero-shot machine unlearning." IEEE Transactions on Information Forensics and Security 18 (2023): 2345-2354.
>
> **RC 4**
>
> We agree that clarifying what constitutes a "correct answer" is critical to understanding the unlearning pipeline.
>
> As we explained in response to ReviewerJzJz, rather than relying on an exact string-level match, our method employs a semantics-aware matching strategy to determine whether the model's output a^\hat{a}a^ matches the correct answer aaa. This is particularly important given the paraphrastic flexibility of LLMs.
>
> Specifically, in Step 3.b of UnStar, we integrate several mechanisms:
> 1. Synonym Matching via WordNet, to accommodate lexical variation (e.g., "poet" vs. "writer").
> 2. Number Normalization, allowing comparisons between numeric and word-form expressions (e.g., "4" ↔︎ "four").
> 3. Abbreviation/Synonym Dictionaries, to resolve domain-specific aliases (e.g., "USA" ↔︎ "United States").
> 4. Fuzzy String Matching (Levenshtein distance via fuzzywuzzy), to permit minor typographic or formatting discrepancies.
> 5. Semantic Similarity Scoring, using Sentence Transformers (e.g., paraphrase-MiniLM-L6-v2), to detect conceptually equivalent responses even with substantial surface-level differences.
>
> This multi-tiered approach ensures robustness against superficial variations while maintaining high semantic fidelity. Only if the generated output passes a liberal but reliable threshold across these criteria do we consider it a correct match. This strategy, discussed previously in our response to Reviewer JzJz, was designed to avoid both false negatives (missing paraphrased correct answers) and false positives (unintentionally treating incorrect outputs as correct).
>
> **RC 5**
>
> We agree that understanding the role of fake justifications (rationales) is central to evaluating our approach. We provide theoretical, and empirical evidence to support our claim that fake rationales improve the unlearning process.
> 1. Theoretical framing: We introduce a reinforcement learning-style policy gradient approximation to formalize our method. We treat the model as a discrete latent variable model where it first samples a rationale and then predicts an answer. This allows us to define separate reward signals for retain and forget data: correct answers on the retain set are rewarded, while incorrect answers on the forget set are rewarded. This selective mechanism enables precise gradient updates guided by rationales.
>
> 2. Empirical evidence: To validate the effectiveness of rationales, we compare against a strong baseline equivalent to the random-label-based method (no rationales). As shown below, UnSTAR significantly outperforms this baseline across all major metrics:
>
> | **Metric**                         | **Yao et al.** | **UnSTAR** |
> |-----------------------------------|----------------|------------|
> | Unlearning Efficacy               | 84             | 100        |
> | Model Utility (retain set accuracy) | 13             | 100        |
> | Response Quality                  | 0              | 92         |
>
> These results underscore the importance of rationales in guiding the model to selectively forget without harming unrelated knowledge.
> In summary, our method offers a theoretically motivated, empirically supported, and interpretable framework for unlearning. We believe that the use of fake rationales is not just a design choice, but a key factor contributing to the effectiveness of UnSTAR—offering a structured alternative to existing approaches such as random label flipping.

---

> > ### Author Response · Authors · 2025-05-04
> > **RC 7**
> >
> > **RC 7**
> >
> > Our evaluation of model utility is not limited to the exact samples seen during fine-tuning on the retain set. While our fine-tuning process does sample from the retain set, it is done efficiently, for each forget-set question, we retrieve a minimum of one semantically relevant question from the retain set and generate multiple paraphrased versions to ensure coverage. This paraphrased augmentation ensures diversity and helps simulate broader generalization.
> >
> > Importantly, our evaluation includes both in-distribution examples from the retain set and out-of-distribution hold-out examples that were never seen during fine-tuning. This is to ensure that the utility metrics reflect the model's ability to generalize beyond the examples it was re-exposed to during the retain-set fine-tuning phase.

---

> > > ### Author Response · Authors · 2025-05-04
> > > **RC 8**
> > >
> > > **RC 8**
> > >
> > > Thank you for this suggestion. We have updated Figure 3 to include a Retain Efficacy curve reflecting model utility. A revised manuscript is uploaded. Figure 3 illustrates the LLM's unlearning and retain efficacy as it progressively unlearns an increasing number of paraphrased versions of the same question. Unlearning efficacy improves monotonically, confirming effective erasure. Retain efficacy experiences a slight decline but recovers steadily thereafter due to our joint optimization strategy. This highlights that our method achieves effective unlearning while preserving, and eventually restoring, performance on the retain set, demonstrating minimal long-term utility degradation.

---

> ### Author Response · Authors · 2025-04-23
>
> **RC 6 and RC 9**
>
> We have now defined the metrics used in our evaluation. The metrics are based on the definitions provided in the EMNLP 2023 paper Liu et al [1].
>
> 1. ROUGE Score: We use the ROUGE-L score, which calculates the longest common subsequence between the ground-truth (GT) and generated answers. Since the GT answers in our dataset are concise, ROUGE evaluates the correctness of the generated answers based on their alignment with the expected response. However, as you pointed out, ROUGE may not fully capture when the model hasn't truly unlearned the desired information, particularly if the generated answer still conveys the same underlying fact despite being worded differently.
>
> 2. GPT Privacy Score: This metric addresses the concern you raised. The GPT Privacy Score evaluates how well the model’s response protects the unlearning target's factual information. For each question, the GT answer, and the model-generated response, GPT-4 rates how much the response still reveals or "leaks" the targeted information, using scores from {1, 2, 3}, where 3 indicates no factual leakage. This means that even if the generated answer uses different phrasing (e.g., "a fictional magic user who defeats Voldemort" instead of "a wizard from Hogwarts"), the GPT Privacy Score will still flag it if it conveys the unlearned fact. A score of 3 signifies that the model has successfully unlearned the target fact and is no longer generating responses that directly or indirectly imply the original information.
>
> 3. GPT Quality Score: In addition to the privacy score, we also assess the GPT Quality Score. This score evaluates the quality of the generated response, considering fluency, relevance, and appropriateness, regardless of correctness. The score ranges from {1, 2, 3}, where 3 denotes fluent, relevant, and appropriate responses.
>
> **To address your question about the privacy score:**
>
> Yes, the GPT Privacy Score is specifically designed to address situations like the one you described. While ROUGE might suggest that the model has unlearned the information (by reporting a high similarity score), the GPT Privacy Score explicitly evaluates whether the model has leaked any of the unlearning target's factual knowledge. If a model response still indirectly implies or references the unlearned information, the Privacy Score will capture this and provide a lower score, indicating that the model has not successfully forgotten the targeted knowledge.
>
> In summary, the GPT Privacy Score is a more robust metric for ensuring that the model has truly unlearned the targeted information, beyond what is captured by traditional evaluation metrics like ROUGE. It directly addresses the challenge of factual leakage, ensuring that the model no longer retains or reveals the knowledge it was intended to forget.
>
> We have added the metrics’ explanation as well as prompts used for evaluation in Appendix A.7 Evaluation Metrics.
>
> [1] Yujian Liu, Yang Zhang, Tommi Jaakkola, and Shiyu Chang. Revisiting who’s harry potter: Towards
> targeted unlearning from a causal intervention perspective. EMNLP 2023
>
> **RC 7 & 8**
>
> We are working this and we will update you shortly.
>
> **RC 9**
>
> Answered along with RC 6 earlier.
>
> **RC 10**
>
> We appreciate the reviewer’s concern regarding generalization across models and datasets. In response, we have added additional results on a new dataset focused on the identity of Spiderman / Peter Parker, in addition to the original Harry Potter forget set. These new evaluations are included in the appendix.
>
> To test robustness, we paraphrased the forget-set queries in multiple forms (e.g., "Can you disclose the real identity of Spiderman?", "Who is the true persona behind Spiderman?") and observed the model’s behavior post-unlearning. After applying our method, the model consistently avoided mentioning Peter Parker, responding instead with generic and deflective answers such as:
> >"The identity of the superhero known as Spiderman is not publicly disclosed."
>
> >"He is commonly recognized as the friendly neighborhood superhero."
>
> This demonstrates that our approach is able to generalize to a different factual identity in a different fictional universe, supporting the claim that our method is not narrowly tailored to a single domain.
>
> ​​In addition to our main experiments on Mistral-7B-Instruct, we have also evaluated UnSTAR on a smaller model, SmolLM-135M-Instruct. UnSTAR was able to effectively erase forget set, e.g., to the query "Where does Harry Potter study?", the response avoids mentioning "Hogwarts" or even providing a direct answer. The results have been added in the Appendix.

---

> ### Author Response · Authors · 2025-04-23
>
> **RC 11**
> We agree that further discussion of limitations can help clarify the scope and applicability of our method. We have added the following points to the discussion section of the paper:
>
> >Limitations. UnSTAR, while effective, comes with a few limitations. First, it assumes the availability of
> question-answer (QA) structured data. While it is possible to extend our method to general free-form text
> (e.g., Wikipedia pages) by extracting pseudo-QA pairs using auxiliary models, such conversions may not
> always faithfully represent the knowledge to be unlearned.
> Second, our approach is best suited to models that support intermediate reasoning, such as those fine-tuned
> with chain-of-thought prompts. These models allow us to intervene not just on final answers but also on
> the rationale behind them. In contrast, associative models like GPT-J, which do not exhibit structured
> reasoning, are less amenable to UnSTAR’s anti-sample based intervention.
> Finally, we note a subtle challenge: even if the final answer is unlearned, it is still possible for a model to
> reveal forgotten information through intermediate reasoning steps. For example, chain-of-thought responses
> may partially reconstruct or hint at unlearned facts. While our method attempts to counteract this by also
> training on misleading rationales, completely eliminating such leakage remains an open problem

---

### Review · Reviewer_Jzjz · 2025-04-09

**Summary Of Contributions:**

The paper introduces a novel targeted unlearning framework for LLMs that uses anti-samples generated via self-taught reasoning. Instead of retraining or merely reversing gradients, UnStar creates misleading rationales and paraphrased questions paired with incorrect answers to selectively “forget” specific associations while retaining other related facts. The approach is iterative, incorporating semantic filtering and a reinforcement learning–inspired training objective. The approach is evaluated across several datasets/benchmarks to demonstrate its efficacy compared to baselines.

**Audience:**

Yes

**Broader Impact Concerns:**

Although the targeted unlearning method aims to remove specific data associations, incomplete unlearning or overfitting to wrong answers could leave residual sensitive information.
Also, reinforcement on the retain set might skew the model’s output, leading to over-memorization of certain patterns and potentially reinforcing biases.

**Claims And Evidence:**

No

**Requested Changes:**

1. The authors should discuss whether introducing and reinforcing incorrect associations might lead to hallucination or overfitting on wrong answers, and analyze its impact on the diversity and general utility of generated responses.
2. Please provide a comparison with random-label based methods, emphasizing how the reasoning-guided anti-sample generation is fundamentally different and potentially more effective.
3. In step 3.b of UnStar, the procedure checks if the model’s output (â) equals the original correct answer (a), can you explain how this is assessed? Like through a word-by-word match or via semantic similarity metrics?
4. In the introduction, it would be helpful to include a brief description of how data samples, learning methods, and loss functions differ in the context of learning versus unlearning, to improve clarity for the reader.

**Strengths And Weaknesses:**

Strengths:
1. The idea of guiding unlearning through generated anti-samples is original, and the approach seems to achieve more fine-grained control on unlearning.
2. The experimental evaluation is detailed and extensive with multiple baselines and benchmarks.

Weakness:
1. The method provides intentionally incorrect answers (with justifications) raises concerns about potential hallucinations. This process may inadvertently lead to overfitting the model to the wrong answers and even create new, unintended associations that could affect answer diversity. The paper does mention filtering to remove “near-correct” wrong answers, but does not fully explore whether such overfitting could harm overall model utility.

2. While the method is conceptually distinct through its use of reasoning-driven anti-sample generation, it would be better if the paper could differentiate itself from simpler random-labeling techniques.

3. By reinforcing correct answers for the retain set, there is a risk that the model might over-memorize these examples, potentially creating new privacy issues. This aspect is not thoroughly addressed.

---

> ### Author Response · Authors · 2025-04-23
>
> **Request Change (RC) 1**
>
> We appreciate this thoughtful observation. The anti-sample generation can be tailored to different unlearning needs and does not inherently promote hallucination or misinformation. For instance, given a forget query like “Where did Harry Potter study?”, there are multiple valid unlearning strategies:
>
> 1. Generalized, non-informative answers such as “He studied at a magical university” offer a safe yet vague response. These are unlikely to introduce hallucinations and are often used to preserve plausibility while avoiding specific facts. However, outright refusals can be jarring or unhelpful for users, making this type of response more desirable in some cases.
>
> 2. Factually incorrect (hallucinated) answers, like “He studied at Ilvermorny”, may be more appropriate in adversarial scenarios—e.g., when a malicious user repeatedly probes for sensitive information such as credit card numbers. In such contexts, providing misleading but plausible falsehoods can act as a defense mechanism, breaking the interaction loop and preventing further exploitation.
>
> This is analogous to social engineering countermeasures in real-world systems. Just as a security agent might give vague or deliberately incorrect directions when someone probes restricted access areas, an LLM can use misleading anti-responses to safely disengage or redirect unwanted queries.
>
> UnStar supports both unlearning directions: the generation of non-informative or misleading anti-samples. These are deliberately controlled to serve specific unlearning purposes and do not inherently result in global hallucination or model-wide misinformation.
>
> To quantify the potential side-effects, we evaluate the model’s performance on the retain set and observe that its utility remains stable. This confirms that unlearning via anti-samples can be done without degrading the model’s overall capabilities.
>
> **RC 2**
>
> Thank you for the valuable feedback. We agree that contrasting UnSTAR with random-labeling approaches (Yao et al. [1]) would help highlight the novelty and effectiveness of our method. Below, we clarify the conceptual differences and back them with empirical results:
>
> **Conceptual Differences:**
>
> Random-label-based unlearning typically involves replacing correct answers with incorrect ones sampled arbitrarily or drawn from a fixed pool of wrong answers. These replacements are often context-agnostic and do not attempt to maintain coherence with the question or rationale. As a result, such anti-samples may not align well with the model’s internal reasoning pathways, reducing their effectiveness in weakening the specific associations to be forgotten.
>
> In contrast, UnSTAR’s reasoning-guided anti-sample generation:
> 1. Produces semantically plausible yet incorrect rationales, ensuring that the model engages with misleading logic rather than simply discarding the sample as noise.
>
> 2. Leverages self-taught reasoning to generate answers and rationales that are structurally similar to the original ones, increasing their effectiveness as adversarial counterexamples.
>
> 3. Applies semantic filtering to ensure the anti-samples are maximally distinct from the correct answer without being nonsensical, which reduces the chance of merely confusing the model.
> This structured misdirection is a key differentiator—by embedding incorrect associations in a format the model “respects” (i.e., coherent reasoning chains), UnSTAR induces forgetting more reliably and with finer granularity.
>
> **Empirical Comparison:**
>
> As shown in the table below, UnSTAR significantly outperforms the random-label-based method Yao et al. [\cite{yaolarge}] across almost all metrics:
> | **Metric**                         | **Yao et al.** | **UnSTAR** |
> |-----------------------------------|----------------|------------|
> | Unlearning Efficacy               | 84             | 100        |
> | Model Utility (retain set accuracy) | 13             | 100        |
> | Response Quality                  | 0              | 92         |
>
> Notably, this method suffers from poor utility and response quality, indicating that while the model forgets the target association, it does so at the cost of general performance. In contrast, UnSTAR achieves complete unlearning without sacrificing utility, highlighting the superior balance achieved by our reasoning-based approach.
>
> [1] Yuanshun Yao, Xiaojun Xu, and Yang Liu. Large language model unlearning. In Socially Responsible Language Modelling Research. NeurIPS Workshop 2024.

---

> ### Author Response · Authors · 2025-04-23
>
> **RC 3**
>
> In Step 3.b of our method, we do not rely on a strict string match between the model’s output a hat and the correct answer a. Instead, we adopt a flexible, semantics-aware matching strategy which integrates multiple comparison mechanisms to ensure robustness:
> 1. Synonym Matching via WordNet to account for lexical variability (e.g., "poet" vs. "writer").
> 2. Word-to-Number Conversion (e.g., "four" ↔︎ 4) for numerically expressed answers.
> 3. Custom Abbreviation/Synonym Dictionaries for domain-specific variations (e.g., "USA" ↔︎ "United States").
> 4. Fuzzy Matching using Levenshtein similarity (via fuzzywuzzy) to allow minor spelling variations.
> 5. Semantic Similarity using Sentence Transformers (specifically, paraphrase-MiniLM-L6-v2) to ensure conceptual alignment even when the phrasing varies significantly.
>
> Only when this multi-pronged matching mechanism determines that the generated output contains a variant of the correct answer (within a liberal but reliable threshold) do we consider it a match and proceed with unlearning interventions.
> This approach ensures that semantic equivalence, rather than surface-level identity, guides the unlearning decision—crucial for robustness when working with large language models that exhibit paraphrastic variability.
>
> **RC 4**
>
> We have revised the introduction to include a clear, structured explanation of how data samples, learning methods, and loss functions operate in the learning setting, and how their counterparts—anti-data samples, unlearning methods, and reversed loss functions—function in the context of unlearning. This clarification improves conceptual alignment and makes the motivation behind our anti-sample-based approach more accessible to the reader. The updated paragraph beginning with “Ways to unlearn?” in the Introduction section is reproduced here for your convenience:
>
> >Ways to unlearn? Machine learning models improve accuracy through training by leveraging three key components: data samples, learning methods, and loss functions. Data samples provide correct input-output mappings (e.g., a question and its true answer), learning methods like gradient descent iteratively adjust model parameters to minimize error, and loss functions (e.g., cross-entropy) penalize incorrect predictions to reinforce accurate associations. In contrast, unlearning aims to reverse or negate specific knowledge. Here, anti-data samples can be crafted to contradict or disrupt previously learned facts, unlearning methods can be adjusted for the model to selectively erase unwanted information, and modified loss functions may promote higher entropy, reduce confidence, or even penalize correct predictions for the forget set. This structured reversal—flipping the semantics of data, tweaking the optimization trajectory, and redefining the objective—forms the foundation of unlearning. While much attention has been given to unlearning methods (Bourtoule et al. (2021); Chundawat et al. (2023a); Sinha et al. (2023)) and the manipulation of loss functions to reverse learning (You et al. (2024); Sinha et al. (2024)), the potential of anti-samples remains largely untapped. Although token-based anti-samples have been previously introduced (e.g., Whp Eldan & Russinovich (2023)), the use of reasoning-driven anti-samples—constructed with falsified answers and misleading rationales—remains novel. Our work is the first to integrate such structured anti-sample generation with reasoning-based unlearning in LLMs. This paper aims to fill that gap."

---

> > ### Comment · Reviewer_Jzjz · 2025-05-02
> >
> > Thanks for your response!
> >
> > Re: RC3, Could you describe how you integrate the five matching checks into a single decision? Specifically, are they applied sequentially or in parallel, how you aggregate or weight their outputs, and resolve cases where some modules agree and others don’t?

---

> > > ### Author Response · Authors · 2025-05-03
> > > **Thank you!**
> > >
> > > Thank you for your response and follow up question. We run five checks in order, once per keyword‑token pair, and stop at the first match. That single “first‑match wins” rule is the aggregation rule we follow. We prepare the following Table to describe the matching mechanism.
> > >
> > > | **Step**                     | **What we do**                                                                                                                 | **Why**                                                         |
> > > | ---------------------------- | ------------------------------------------------------------------------------------------------------------------------------ | --------------------------------------------------------------- |
> > > | **1. Pre‑clean**             | strip punctuation → lower‑case                                                                                                 | avoid trivial mismatches                                        |
> > > | **2. Split**                 | gold answer ⇒ **keywords**  k₁ … k\_m.  model output ⇒ **tokens**  w₁ … w\_n                                                  | lets us compare one word at a time                              |
> > > | **3. Compare each (kᵢ, wⱼ)** | run the **five rules in a fixed order** (*exact → abbreviations → WordNet → fuzzy → semantic*) and **stop as soon as one fires** | keeps precision high as looser rules can’t override a stricter match |
> > > | **4. Record a match**            | if any rule returns True for a pair, mark kᵢ as “matched” and move on to the next keyword                                             | simple bookkeeping                                              |
> > > | **5. Decide “match?”**     | if **≥ 1 keyword** matched → output counted as "match"; otherwise "no match"                                                    | liberal enough for paraphrases, but still reliable              |
> > >
> > >
> > > **Why this works?** A strict rule firing stops the cascade → no disagreement possible. If a strict rule doesn’t fire, the cascade simply moves on; the first rule to fire decides a "match". Different keywords are checked independently, so a loose rule can still rescue a paraphrased keyword that the strict rules missed.

---

> ### Comment · Reviewer_Jzjz · 2025-05-05
>
> Thank you for clarifying it. I remain uncertain about its robustness in practice, but I recognize that this is an open challenge and doesn’t take away the paper’s core contributions. I would still suggest including this detailed description of how the checks are integrated and some empirical evidence of their effectiveness in a later revision.

---

### Decision · Action_Editor_VfaV · 2025-05-14

**Recommendation:** Accept as is

**Comment:**

This paper studies unlearning in LLMs in the context of question-answering (QA). The authors propose a method, called UnSTaR that is an iterative approach for unlearning a forget set of QA pairs based on finetuning the model on "anti-samples" (misleading incorrect responses). They generate paraphrases of the question in each forget-set QA pair, and pair each paraphrase with an incorrect response, and then finetune on this dataset. They also do light-weight finetuning on the retain set (QA pairs that don't need to be forgotten) to maintain model utility.

The reviewers pointed out that the method is "intuitive and straightforward, likely to be replicable and form a solid baseline for future works" (Reviewer q1SN), "it is intuitively clear why it works" (Reviewer 8JfS). They also pointed out that the method performs strongly: "The experimental evaluation is detailed and extensive" (Reviewer Jzjz), "The results suggest strong performance across all metrics" (Reviewer q1SN), "UnStar's empirical performance is quite strong, and the experiments are quite extensive" (Reviewer 8JfS).

During the rebuttal, the authors addressed weaknesses pointed out by the reviewers. They discussed whether the use of anti-samples can lead to hallucinated responses for retained data or other unwanted side-effects. They corrected unsupported claims (as discussed above), expanded discussion of related work significantly, clarified how they quantify what constitutes a "correct answer" (an issue brought up by two reviewers), they discussed the metrics at length (as Reviewer q1SN pointed out, ROUGE is insufficient to measure unlearning, as it relies on exact substrings, the authors clarified the motivation for using GPT metrics to address this), also clarifying other aspects of evaluation like the use of held-out examples. The authors add results on a new dataset too to address a concern of insufficient evidence of generalization across models and datasets. The authors have also discussed comparison of their method against retraining in terms of the computational cost, and added a discussion of other limitations. These changes made in the rebuttal clarify important concerns and strengthen the paper.

Overall, I recommend acceptance as this work makes a valuable contribution to the unlearning literature, providing an intuitive straightforward method that is effective against baselines as shown in extensive experiments where strong empirical performance is observed.

In the camera ready version:

For the methods using random-label-based unlearning in the vision classification literature, please cite:
- (i) Amnesiac Machine Learning. Graves et al. 2020 -- this is the first work, to the best of my knowledge, that proposed the idea of random labeling for unlearning.
- (ii) Boundary Unlearning: Rapid Forgetting of Deep Networks via Shifting the Decision Boundary. Chen et al. CVPR 2023 -- in their "boundary shrinking" variation, instead of assigning a random label, they assign the label of a similar image that belongs to a different class. This may be more analogous to the proposed procedure in this paper that looks for "misleading" incorrect answers.

Please also incorporate clarifications made during the rebuttal, in particular when it comes to evaluation metrics.

**Audience:**

This paper is on the important topic of unlearning in LLMs and is of interest to the TMLR community.

**Claims And Evidence:**

This paper studies unlearning in LLMs in the context of question-answering (QA). The authors propose a method, called UnSTaR that is an iterative approach for unlearning a forget set of QA pairs based on finetuning the model on "anti-samples" (misleading incorrect responses). They generate paraphrases of the question in each forget-set QA pair, and pair each paraphrase with an incorrect response, and then finetune on this dataset. They also do light-weight finetuning on the retain set (QA pairs that don't need to be forgotten) to maintain model utility.

The reviewers pointed out issues with some claims, namely that this work is the first to do "anti-sample unlearning". As the reviewers correctly pointed out, related approaches have been used before in LLMs (in the WHP method cited by the authors), as well as in the vision classification literature (where a random label is assigned to forget-set images, and the model is then finetuned with those incorrect labels). The authors responded to this feedback by establishing a distinction of those works and theirs by claiming that they use "reasoning-driven anti-samples" that are constructed with falsified answers and misleading rationales. This seems reasonable and novel. The authors have correctly cited and discussed the other works brought up by the reviewers using anti-samples in the revision.  Please see my comment below for additional citations that are needed. Reviewer q1SN brought up another claim that was unsupported, in relation to prior work, but the authors have addressed that in the revision too, and expanded the related work section significantly. Finally, during the rebuttal, authors provided additional empirical evidence to support our claim that fake rationales improve the unlearning process.

Aside from that, the reviewers noted that "The experimental evaluation is detailed and extensive" (Reviewer Jzjz), "The results suggest strong performance across all metrics" (Reviewer q1SN), "UnStar's empirical performance is quite strong, and the experiments are quite extensive" (Reviewer 8JfS), supporting the claim that this method outperforms prior baselines.